# Emergency department visits among patients transported by law enforcement officers

**David L. Rosen**[1]*, **Debbie Travers**[2]

**1** Division of Infectious Diseases, School of Medicine, University of North Carolina at Chapel Hill, Carrboro, North Carolina, United States of America, **2** School of Nursing, University of North Carolina at Chapel Hill, Chapel Hill, North Carolina, United States of America

* drosen@med.unc.edu

**Data Availability Statement:** The data underlying this study, which were provided by the NC Division of Public Health, contain both sensitive and identifying information. Per DLR's data use agreement with the Division of Public Health, we

## Abstract

Law enforcement officers frequently encounter people with health conditions. We sought to estimate the rates, diagnoses, and characteristics of emergency department (ED) visits among patients transported directly by law enforcement. We analyzed statewide North Carolina Emergency Department data for adults, aged 18+ years, from 2009 to 2016. We estimated transport rates using census data; categorized primary ED diagnoses into 13 major and 8 substituent categories; compared county transport rates by rurality; and examined patient characteristics. There were 136,240 patients transported by law enforcement; annual rates increased from 186.9 (per 100,000 adult residents) in 2009 to 279.2 in 2016. Among visits, 67.7% were among men, the median age was 37 years, and 20.4% resulted in a hospital admission. Most common primary diagnoses were Mental Health Diagnoses (43.1%)—including Schizophrenia and other Psychotic Disorders (7.6%), Mood Disorders (9.7%), and Alcohol and Substance Use (10.7%)—followed by Injury and Poisoning (12.4%) and Circulatory conditions (4.1%). Involuntary commitments constituted 22% of all visits. The median transport rate in rural counties, 291.1, was 2 times that of large metro counties, 145.1. The visit rate increased by nearly 50% during the study period, with the highest rates in rural counties. Many transports were for Mental Illness and involuntary commitments. The relatively common occurrence of law enforcement transports suggests the need for greater research to understand factors influencing law enforcement transport decisions, the impact of these transports on patient health and safety, and the repercussions on patient care of a growing officer presence in EDs.

## Introduction

US law enforcement officers have tens of millions of interactions with the public annually. More than one in every five persons aged 16 years or older, or about 53 million people, has at least one encounter with a law enforcement officer annually [1]. Officers can respond to accidents [1] and other medical emergencies, and are often the first to arrive at scenes requiring medical or behavioral health intervention [2].

When law enforcement encounters require medical care, Emergency Management Systems (EMS) ambulances typically transport patients to nearby emergency departments (EDs), with

are legally restricted from sharing the study data. However, data can be requested here: https://ncdetect.org/.

**Funding:** DR, DT, 1R01MD012469-01, National Institute on Minority Health and Health Disparities of the National Institutes of Health, https://www.nimhd.nih.gov/. The funders had no role in study design, data collection and analysis, decision to publish, or preparation of the manuscript.

**Competing interests:** The authors have declared that no competing interests exist.

law enforcement accompanying patients who are in custody [3]. Although less well appreciated, officers may themselves transport patients to an ED when the need for care is urgent [4]; in instances when ambulances are temporarily unavailable; or when direct transport is simply most expeditious. The decision for law enforcement rather than EMS to conduct ED transports is—with one exception—not guided by state policy. Rather the decision for law enforcement to transport is based on the particular situation or local decisions [5–9], as in in Philadelphia, where police are explicitly allowed to transport blunt trauma patients directly to the hospital [10].

The exception, ascribed in state laws across the US, is that officers are typically responsible for ED transports pursuant to involuntary commitment (IVC) orders, in which courts mandate that individuals deemed to be at risk of harming themselves or others are taken into custody for psychiatric evaluation [11]. Law enforcement officers' responses to IVCs—and, more generally, their interactions with those experiencing mental health crises—have long raised questions about how best to intervene in these situations. These questions have intensified in the wake of a national discussion prompted by the police mediated killings of Black persons, including that of Daniel Prude, who died as the result of law enforcement restraint techniques that were applied while he was having a psychotic episode. These discussions have been enveloped into the larger debate about the appropriate roles and responsibilities of law enforcement to engage in activities beyond those directly addressing criminal activity.

Nevertheless, the extent to which law enforcement officers engage in direct transports to EDs is unknown. A more global understanding of these transports could inform law enforcement officers' training needs as well as the need for medical and mental health resources in their respective communities. More broadly, such information could help inform the debate regarding the appropriate breadth of law enforcement agencies' responsibilities. In response, we used statewide North Carolina emergency department data from an eight-year period to examine the prevalence and nature of non-ambulance law enforcement transports to emergency departments.

## Methods

### Setting

In NC, there are 504 law enforcement agencies [12]. In 2008 and 2015, the two most recent years with data, there were respectively an estimated 23,442 [12], and 21,143 [13] fulltime sworn law enforcement officers in the state.

### Data sources

For the years 2009–2016, we obtained visit-level records from a statewide emergency department surveillance database, the NC Disease Event Tracking and Epidemiologic Collection Tool (NC DETECT), which is administered in a collaboration between the [Institution-blinded] and the NC Division of Public Health. As described below, we obtained only those records corresponding to ED transports by law enforcement. At the time of our data request, 2016 was the most recent year with available data. NC DETECT records represent greater than 99.5% of all ED visits from acute-care, civilian, hospital-associated EDs in the state [14]. Medical coding occurs in each hospital. When hospitals use different coding systems for key variables, standardization of those variables is conducted using the Data Elements for Emergency Department Systems; additional coding details are described elsewhere [15]. We also obtained US census estimates for NC counties for years 20009–2016, inclusive [16].

## Data elements

Variables in the NC DETECT ED data included patient sex, age, and county of residence; transport mode; chief complaint; International Classification of Diseases (ICD) diagnosis codes; expected payment source; and ED discharge disposition. Variables for race/ethnicity were not collected by NC DETECT.

## Study population

Our study population included all NC ED patients for the years 2009–2016 who were aged 18 years or older at the time of their ED visit and had a transport code of "Walk-in following non-ambulance, law enforcement transport," meaning that law enforcement transported a patient to an ED without the use of an ambulance.

## Analysis

For the entire study period and for each year, we estimated the rate (per 100,000 adult state residents) of ED visits resulting from law enforcement transport. Using clinical categories and IVC-related transports described below, we also estimated annual transport rates among those with and without diagnoses indicating Mental Illness, and we estimated the transport rate for IVCs. We also estimated transport rates stratified by county. In exploratory analyses to examine differences in transport rates by county rurality, we collapsed six classifications from the National Center for Health Statistics [17] to categorize patients' county of residence into: Rural, Small to Medium Metro, and Large Metro. We then calculated the median transport rate for each rurality category and used the Wilcoxan Rank Sum test to compare distributions of transport rates across categories.

We examined the distribution of all continuous and categorical variables in the ED data relating to patient and visit characteristics. We used the Agency for Healthcare Research and Quality's Clinical Classification System (CCS), which categorizes all International Classification of Disease (ICD) diagnoses codes into 18 major clinical categories and several hundred substituent categories [18]. Versions of the CCS have been created for both ICD versions 9 and 10, facilitating examination of clinical categories across the two ICD versions. Corresponding with the timing of a switch in NC EDs from using ICD-9 to using ICD-10, we used CCS for ICD-9 to code diagnoses generated through September 31, 2015, and we used the CCS for ICD-10 to code diagnoses through the remainder of the study period (October 1, 2015 to December 31, 2016). We estimated the prevalence of the 18 major CCS categories, although in reporting our results, collapsed six of the CCS categories into a single category because these health conditions were infrequent and of modest substantive interest. We also reported the prevalence estimates for a total of 10 substituent categories, which fell within the major CCS categories "Mental Illness" and "Injury and Poisoning." For a listing of the 13 major and 10 substituent categories, please see Table 2. In addition to estimating the prevalence of each of our 13 major and 10 substituent clinical categories for the entire study period, we also estimated prevalence estimates for each of the two time periods defined by the use of different ICD versions. Congruent with ICD coding guidelines, we interpreted the first listed diagnosis as the one "chiefly responsible" for the ED visit [19]; we focused on this diagnosis for our analyses, referring to it as patients' "primary diagnosis."

In addition to examining disease prevalence by clinical category, two investigators reviewed the 1000 most common free-text chief complaints. These 1000 most common chief complaints comprised the chief complaint in 71% of all observations. This review suggested that IVCs were a particularly common reason for transport, but not necessarily reflected in patient's diagnoses. In reviewing the 1000 most common chief complaints, we identified all text variations indicative of IVCs; these included the terms "ivc", "invl", "invol" and "inv commit." We

developed a case insensitive query based on the above terms to identify IVC-related transports. We then reviewed the chief complaints identified by our query to verify that they reflected IVCs, and we created a dichotomous variable for IVCs (yes/no), to estimate the prevalence and annual rate of IVC transports. Because IVCs were identified in the chief complaint variable and mental illness was identified using the variable for primary diagnosis, our estimates of transport rates by IVC and by Mental Illness were not mutually exclusive.

Finally, in ancillary analyses we examined the age and sex distribution by rurality for all law enforcement transports and for selected causes: mental health condition, IVC, overdose, and wounds.

All analyses were conducted at the visit-level as identifiers were not available to determine which, if any, ED visits represented the same patient. Analyses were conducted using SAS version 9.4 (Cary, NC).

This study was approved by the University of North Carolina at Chapel Hill Institutional Review Board Approval #15–0609 and the North Carolina Department of Public Health.

## Results

### Annual trends and geographic distribution of law enforcement ED transports

From January 1, 2009 to December 31, 2016 there were 136,240 ED visits resulting from non-ambulance law enforcement transport. The annual number of visits increased from 2009 (n = 13,412) to 2012 (n = 17,938), declined in 2013 (n = 16,148), and increased through 2016 (n = 21,949) for an increase from 2009 to 2016 of 61%. The law enforcement transport rate for the state was 226 per 100,000 residents; similar to the annual number of visits, the rate grew 49% across the study period, from 186 to 279 per 100,000 (Fig 1). For the year 2016, law enforcement transported patients to the ED, on average, 60 times every day.

Across the entire study period, the transport rate by county ranged from 28 to 977 per 100,000. Using our measure of rurality, the median transport rate (per 100,000) was 145.1 in Large Metro counties (n = 12), 214.8 in Small to Medium Metro counties (n = 34), and 291.1 in Rural counties (n = 54) (Fig 2). The differences in the distribution of rates between Rural counties and both Small to Medium counties (p < 0.05) and Large Metro counties (p < 0.05), were statistically significant, but the difference in the distributions of rates for Small to Medium counties compared to those of Large Metro counties were not statistically significant (p = 0.26).

### Annual trends of law enforcement ED transports for all mental health causes and IVC

Across the study period, the rate of mental health-related transports grew from 75.9 to 134.2 per 100,000, an increase of 77%, while the rate of non-mental health-related transports grew from 111.0 to 145.0 per 100,000, an increase of 31%. The rate of transports specifically for IVC grew from 43.9 to 69.9 per 100,000, an increase of 59%, with most of this increase occurring from 2015 to 2016 (Fig 1). Across the study period, IVC accounted for 22.6% (30,800/136,240) of all law enforcement ED transports and 37% (21,633/58,694) of all law enforcement ED transports for which the primary diagnosis was a mental health condition; conversely, 70% (21,633/30,800) of all IVC transports had a mental health condition listed as the primary diagnosis.

### Characteristics of ED visits among patients transported by law enforcement

About two-thirds of the visits were among men (68%) and the median age was 37 years (interquartile range: 27–50). Twenty percent of patients were admitted (including 6% to Psychiatry)

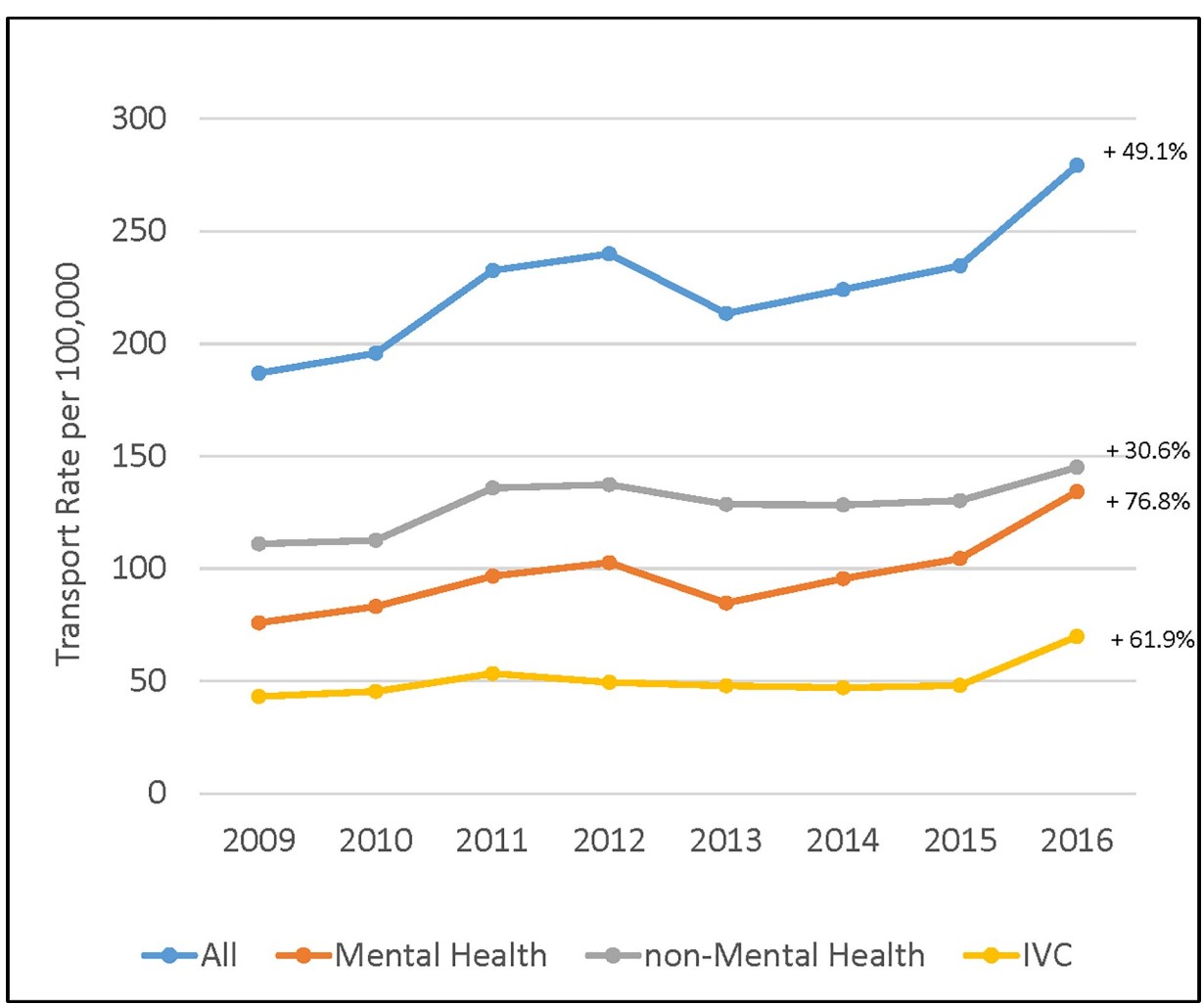

**Fig 1. Annual rates of law enforcement transports to E.D.s by mental health status and Involuntary Commitment (IVC), North Carolina, 2009–2016.** Note: mental health and involuntary commitment (IVC) rates are not mutually exclusive; percent change in rate from 2009 to 2016 is presented to the right of each trendline.

for an inpatient stay, 54% were discharged from the ED, 13% were transferred, 2% left against or without medical advice, and <1% died. Half of all deaths (52/104) were among patients transported for IVC, although all but one IVC death occurred from 2009–2012. Greater than 70% (n = 31) of the IVC deaths had a primary diagnosis of Schizophrenia/psychotic disorders or mood disorder; 20% (n = 8) had a primary diagnosis of substance use.

Among all ED visits, patients most commonly had government-sponsored insurance (39%) —including Medicaid (17%), Medicare (17%), and "other" government payments (4%)—followed by self-pay (27%) and private insurance (18%) (Table 1).

### Diagnoses among ED patients transported by law-enforcement

The most common primary diagnoses at ED disposition were for Mental Illness (43%) and Injury and Poisoning [commonly referred to as "Drug Overdose"] (12%) (Table 2). Among those with a Mental Illness diagnosis, the most common subcategories included Mood disorders (10%), Schizophrenia/Psychotic disorders (8%), Depressive disorders (6%),

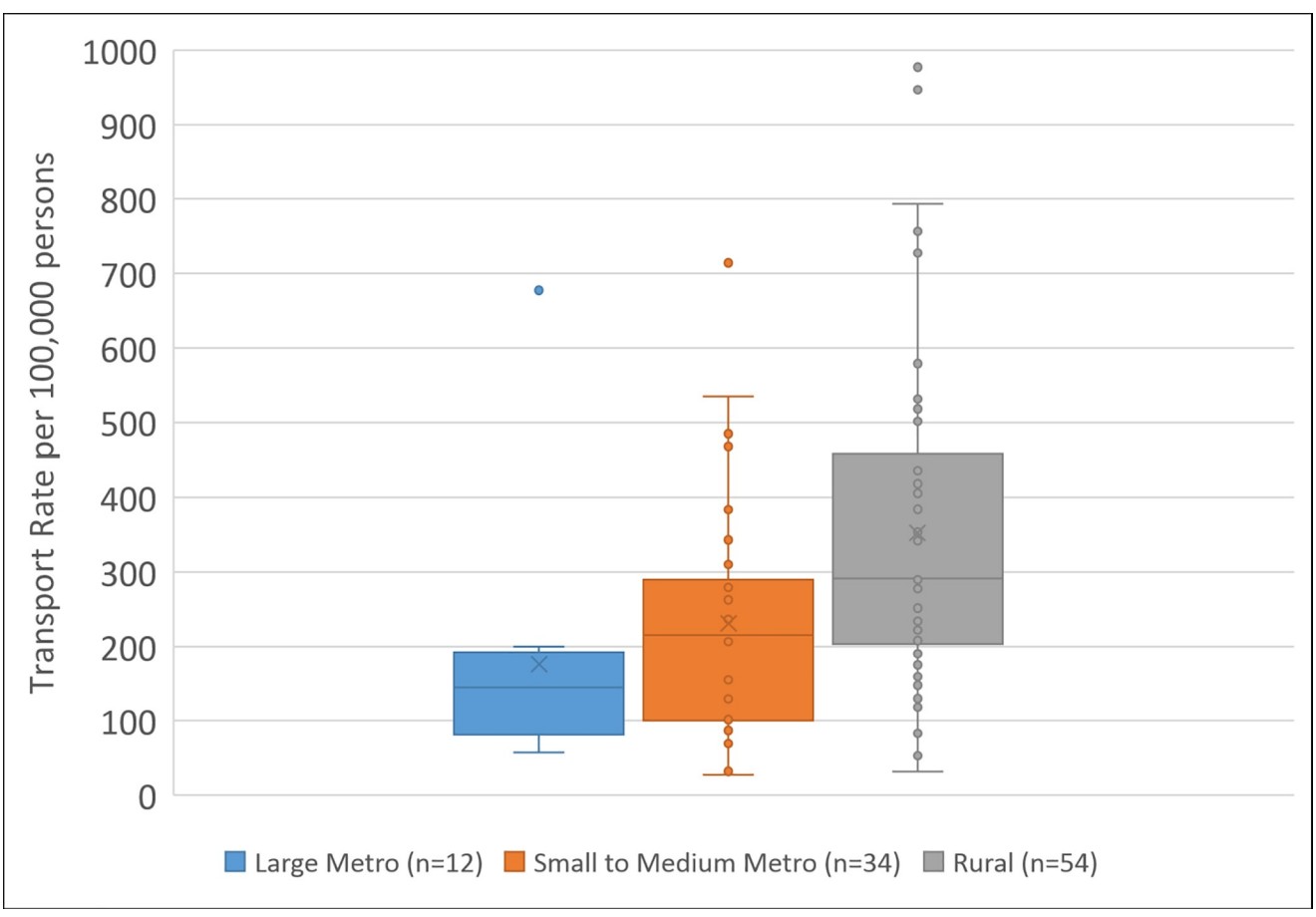

**Fig 2. Law enforcement ED transport rates by county rurality, North Carolina, 2009–2016.** Note: The figure depicts box and whiskers plots. The middle line in the box represents the median; the "X" represents the mean. The upper and lower borders of the box represent the span of interquartile range, and the whiskers extend to the minimum and maximum non-outlier values.

Substance-related disorders (6%), Alcohol-related disorders (5%), Bipolar (4%), and Suicide/self-injury (4%). Within the category Injury and Poisoning, about 7% had either an open wound, superficial injury, or contusion. Less than 1% were diagnosed for Poisoning. Other health conditions including those of the following systems or categories: circulatory (4%), nervous (3%), respiratory (2%), endocrine (3%) and infectious and parasitic (1%), blood (0.5%) and neoplasm (0.2%). The distributions of diagnoses were relatively consistent across the different ICD versions. The most notable difference in shifting from ICD-9 to ICD-10 coding (and the corresponding CCS programs) was a decline in conditions classified as "Injury and Poisoning" and an increase in conditions classified as "Substance related disorders" (S1 Table).

In ancillary analyses we found that the age and sex distributions among all patients transported by law enforcement differed from those of the general population, but were generally similar to the age and sex distribution of patients transported by law enforcement specifically for mental illness, IVC, and Overdose; compared to these latter conditions, those transported for Wounds had a higher proportion of men. For each of these conditions, differences in age and sex distributions by rurality were modest (S2 Table).

**Table 1. Characteristics of adult Emergency Department (ED) patients transported by law enforcement, North Carolina, 2009–2016.**

| | | N | % |
|---|---|---:|---:|
| Age | | | |
| | 18–26 | 32381 | 23.8 |
| | 27–49 | 68781 | 50.5 |
| | 50+ | 35078 | 25.7 |
| Gender | | | |
| | Male | 92178 | 67.7 |
| | Female | 44052 | 32.3 |
| | Missing/unknown | 10 | 0.0 |
| Insurance | | | |
| | Private (Insurance company) | 24308 | 17.8 |
| | Self-pay | 37266 | 27.4 |
| | Medicaid | 23690 | 17.4 |
| | Medicare | 23592 | 17.3 |
| | Other government payments | 5731 | 4.2 |
| | Workers compensation | 1589 | 1.2 |
| | Other | 15697 | 11.5 |
| | No charge | 128 | 0.1 |
| | Missing/unknown | 4239 | 3.1 |
| ED Disposition | | | |
| | Admitted (any) | 27784 | 20.4 |
| | To ICU | 346 | 0.3 |
| | To Psychiatry | 8749 | 6.4 |
| | To another unit | 18689 | 13.7 |
| | Observation | 574 | 0.4 |
| | Transferred | 17602 | 12.9 |
| | Discharged | 73439 | 53.9 |
| | Left Against Medical Advice/with advice/without advice | 2706 | 2.0 |
| | Other | 4073 | 3.0 |
| | Died | 104 | 0.1 |
| | Missing/unknown | 9958 | 7.3 |

## Discussion

There is increasing interest in both the interaction of law enforcement with the public as well as the health of people in the criminal justice system. In this analysis, which to our knowledge is the first of its kind, we used statewide data to examine the extent that law enforcement officers themselves transport people to EDs and the underlying characteristics and health conditions of the transported patients.

In 2009, there were over four million visits to NC EDs—including more than 550,000 by ambulance—suggesting that law enforcement transports comprise a small proportion of all ED visits [15]. Nevertheless, we found that the rate of law enforcement transports increased nearly 50% from 2009 to 2016, while across a similar interval, the number of law enforcement officers in the state grew by only 10%, suggesting that for the average officer, ED transports during the study period were increasingly common. Moreover, 1 in 5 transported patients was admitted to the hospital and more than 100 patients died, suggesting that people with high levels of medical need were included among those transported by law enforcement.

**Table 2. Disease prevalence among adult emergency department patients transported by law enforcement, North Carolina, 2009–2016.**

| | | N | % | L95% CI | U95% CI |
|---|---|---|---|---|---|
| Mental Illness | | 58694 | 43.1 | 42.8 | 43.4 |
| | Schizophrenia and other psychotic disorders | 10302 | 7.6 | 7.4 | 7.7 |
| | Mood disorders | 13250 | 9.7 | 9.6 | 9.9 |
| | Anxiety disorders | 3959 | 2.9 | 2.8 | 3.0 |
| | Screening and history of mental health and substance abuse | 5917 | 4.3 | 4.2 | 4.5 |
| | Alcohol-related disorders | 7082 | 5.2 | 5.1 | 5.3 |
| | Substance-related disorders | 7425 | 5.5 | 5.3 | 5.6 |
| | Suicide and intentional self-inflicted injury | 5176 | 3.8 | 3.7 | 3.9 |
| Injury and poisoning | | 16931 | 12.4 | 12.3 | 12.6 |
| | Open wounds | 4889 | 3.6 | 3.5 | 3.7 |
| | Superficial injury; contusion | 3764 | 2.8 | 2.7 | 2.9 |
| | Poisoning | 943 | 0.7 | 0.7 | 0.7 |
| Diseases groups 10–15* | | 7435 | 5.5 | 5.3 | 5.6 |
| Diseases of the blood and blood-forming organs | | 729 | 0.5 | 0.5 | 0.6 |
| Diseases of the circulatory system | | 5595 | 4.1 | 4.0 | 4.2 |
| Diseases of the digestive system | | 2348 | 1.7 | 1.7 | 1.8 |
| Diseases of the nervous system and sense organs | | 3503 | 2.6 | 2.5 | 2.7 |
| Diseases of the respiratory system | | 2434 | 1.8 | 1.7 | 1.9 |
| Endocrine; nutritional; and metabolic diseases and immunity disorders | | 3559 | 2.6 | 2.5 | 2.7 |
| Infectious and parasitic diseases | | 1190 | 0.9 | 0.8 | 0.9 |
| Neoplasms | | 216 | 0.2 | 0.1 | 0.2 |
| Residual codes; unclassified; all E codes | | 5069 | 3.7 | 3.6 | 3.8 |
| Symptoms; signs; and ill-defined conditions and factors influencing health status | | 13186 | 9.7 | 9.5 | 9.8 |
| Missing | | 15333 | 11.3 | 11.1 | 11.4 |

*AHRQ Clinical Classification Software (CCS) groups 10–15: Diseases of the genitourinary system (10); Complications of pregnancy; childbirth; and the puerperium (11); Diseases of the skin and subcutaneous tissue (12); Diseases of the musculoskeletal system and connective tissue (13); Congenital anomalies (14); Certain conditions originating in the perinatal period (15).

Note: excludes 18 observations that were missing transport month and year.

The large number of transports, on average 60 per day in the last observed year, raise questions about whether officers have sufficient training, access to appropriate medical supplies, and the ability to communicate with local EDs to prepare medical staff for the patient's arrival. Additionally, compared to EMS transport, law enforcement transport may be severely limited in properly immobilizing trauma patients or in providing care during the transport.

With these overarching considerations in mind, we found that IVCs accounted for more than one in every five law enforcement ED transports. Law enforcement's interaction with people living with severe mental illness, including those who have been ordered for IVC, requires a wide range of skills to manage the scene and de-escalate the encounter. To address these difficult encounters, many US law enforcement agencies have adopted the Crisis Intervention Team (CIT) model. CIT, which includes 40 hours of officer training, promotes coordination between 911 dispatch centers, law enforcement and mental health providers, with the goals of keeping scenes safe and diverting persons in mental health crises away from the criminal justice system and into mental health care [20]. CIT has been adopted by more than 2500

law enforcement agencies nationwide [21], although implementation can vary greatly by site [22]. As of 2015, about one-third of all NC law enforcement officers had undergone CIT training [13], but the extent to which CIT-trained officers transported those with behavioral health problems is unknown.

The growing number of mental health visits in our study, including those for IVCs, appear to have contributed to increasing pressure on EDs to serve the state's behavioral health needs. In 2016, the NC Hospital Association reported that 30% or more of emergency department beds were being used by patients waiting to be transferred to a mental health service provider, with average wait times of two to four days [23]. These delays not only consumed valuable ED resources, but they created barriers to appropriate care for people in crisis. In response, NC passed legislation in 2018 enabling a wider range of credentialed professionals to conduct the preliminary psychiatric evaluations to determine if the patient satisfies criteria for hospitalization and further behavioral health evaluation; allows transports by approved entities other than law enforcement; and mandates that regional mental health managed care organizations develop plans for IVC implementation in their respective communities so that patients do not become stranded in EDs [24]. Future evaluations will be needed to determine if this legislation achieves its intended effects in reducing ED wait times and in reducing IVC transports by law enforcement.

In addition to high rates of IVC transports, we found that the distribution of health conditions among patients in our study differed greatly from those published for the general state ED patient population [15]. Compared to published estimates of the general ED population, our population was about five times more likely to have a primary diagnosis of a mental health condition, but less likely to have a primary diagnosis of a chronic problem, such as diabetes or cardiac problems, or a diagnosed injury. Nevertheless, injuries were the second most common diagnosis group among patients transported by law enforcement. Although there is some evidence that patients' survival from serious penetrating wounds is higher with immediate law enforcement transports compared to waiting for EMS [10], these types of transports—a subcomponent of "Open Wounds"—were uncommon in our data. This finding that penetrating wounds account for only a small proportion of all transports, suggests the need for broader studies examining the impact of law enforcement transports on other types of health outcomes.

Overall, the proportion of people transported for drug overdose (i.e. poisoning) in our population (0.7%) was about twice that in the general population. Considering that NC—like much of the country—is experiencing an epidemic of opioid-related deaths with marked increases since 2014 [25], the encounters represented in our data likely represent a fraction of all law enforcement interactions with people who have overdosed. Across the state many law enforcement agencies recently adopted policies for their officers to carry naloxone, the medication used to reverse opioid overdoses, signaling officers increased involvement in the delivery of care, regardless of transport [26].

In addition to considerations of transport urgency, which may prompt law enforcement transports for wounds and for drug overdose, another factor possibly underlying the use of law enforcement for patient transports is rurality. By their very nature, rural areas have populations spread over greater distances and fewer public safety resources, including ambulances, as compared to metro areas. In our data, rates of law enforcement transports were about twice as high in rural areas as compared to metropolitan areas. While the observed higher rates of transport in rural counties likely reflects numerous factors (e.g. car ownership, public transportation, ED travel distances), these findings highlight the need to determine whether these areas have the appropriate EMS and mental health resources to address resident's needs, and these areas could be prioritized in future studies to determine whether law enforcement

officers have sufficient training for and resources to support transports. Such studies could inform the "defund the police" debate, in which there is growing interest in having community groups rather than law enforcement conduct IVCs and other behavioral health interventions.

Law enforcement and ED staff often work collaboratively to secure patient and staff safety. However, conflicting priorities can emerge, particularly around the use of restraints, patient confidentiality, and evidence collection via clinical laboratories or "body cams" [27]. With the growing number of law enforcement transports to EDs, the potential for these tensions is also increasing, suggesting both a need and an opportunity for research to promote best practices for law enforcement and ED medical staff interactions.

This study has a number of limitations. First, information on transportation mode in the ED data is sometimes inferred by a clerk and thus potentially inaccurate. Transportation mode is also commonly missing [15], potentially underestimating law enforcement transports. Nevertheless, we speculate that the presence of law enforcement officers with a patient and officers' communication of patients' security needs to hospital staff likely result in law enforcement transports being more comprehensively and accurately recorded than other transport modes (e.g. walk-ins). Second, in our analysis of rurality, we used patients' county of residence as a surrogate for the county of transport. Third, ED diagnoses may change for patients admitted to the hospital or transferred. Fourth, according to NC DETECT personnel, beginning with implementation of ICD-10 Codes in October 2015, the first listed code is not reliably the primary diagnosis. Nevertheless, in analyses stratified by ICD-Code version (S1 Table), we found general consistency in the prevalence of diagnoses across ICD versions, and note that the increase in mental health-related transports appears to be unrelated to the transition in ICD versions (S1 Fig). Fifth, our estimate of IVC-related transports was based on analysis of free text, and therefore may have excluded instances in which IVC was not clearly documented. Sixth, without a narrative describing the incident prompting the transport, we are unable to determine why law enforcement rather than an ambulance transported the patient. Finally, our results are applicable to North Carolina and may not be generalizable to other states.

In the context of limited guidelines shaping law enforcement medical transports and calls for diminishing law enforcement involvement in non-criminal matters, we found that rates of law enforcement ED transports—particularly for mental health issues—increased substantially over time, suggesting that with growing frequency, officers were responsible for the health of people with whom they contact. Our findings suggest several lines of needed research. These include examining factors influencing the occurrence of these transports such as community resources supporting emergency responses, law enforcement officer training, and IVC policies as well as the impact of implementing alternative models which diminish law enforcements' role in non-criminal responses. Research efforts may be particularly relevant in rural areas considering their high rates of law enforcement transport but more limited healthcare resources. Additionally, research is needed to examine the impact of these transports on tensions between officers and ED medical staff and the effect of law enforcement transport on patient outcomes and on community relations. Taken together, this information could be used to craft more prescriptive policies, beyond those for IVC, which provide guidance for the appropriate use of law enforcement transports.

## Supporting information

**S1 Fig. Number of mental health-related EDs transports by law enforcement, North Carolina, 2009–2016.** The figure demonstrates that the steep increase in MH-related ED transports begins approximately 4 months following the transition from ICD-9 to ICD-10 suggesting that

the increase is not simply an artifact of the transition.
(PDF)

**S1 Table. Disease prevalence among adult emergency department patients transported by law enforcement, North Carolina, 2009–2016, stratified by ICD periods.** *AHRQ Clinical Classification Software (CCS) groups 10–15: Diseases of the genitourinary system (10); Complications of pregnancy; childbirth; and the puerperium (11); Diseases of the skin and subcutaneous tissue (12); Diseases of the musculoskeletal system and connective tissue (13); Congenital anomalies (14); Certain conditions originating in the perinatal period (15). Note: excludes 18 observations that were missing transport month and year.
(XLSX)

**S2 Table. The table provides the distribution of age and gender by rurality in the general state population and for law enforcement transports to an emergency department.** The associations between age group and rurality were statistically significant (p<0.05) for all transports, mental health problems, IVCs, overdoses, and wounds. The associations between gender and rurality were statistically significant for all transports and for wounds, but not for mental health problems, IVCs, or overdoses.
(XLSX)

## Acknowledgments

The North Carolina Disease Event Tracking and Epidemiologic Collection Tool (NC DETECT) is an advanced, statewide public health surveillance system. NC DETECT is funded with federal funds by North Carolina Division of Public Health (NC DPH), Public Health Emergency Preparedness Grant (PHEP), and managed through a collaboration between NC DPH and the University of North Carolina at Chapel Hill Department of Emergency Medicine's Carolina Center for Health Informatics (UNC CCHI). The NC Office of Emergency Medical Services (NC OEMS) and NC DETECT Data Oversight Committees do not take responsibility for the scientific validity or accuracy of methodology, results, statistical analyses, or conclusions presented.

The authors would like to thank Elena DiRosa for assistance with manuscript preparation.

## Author Contributions

**Conceptualization:** David L. Rosen.

**Formal analysis:** David L. Rosen.

**Writing – original draft:** David L. Rosen, Debbie Travers.

**Writing – review & editing:** David L. Rosen, Debbie Travers.

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
