## [Decision Letter · Decision Letter 0]

21 Jul 2020

PONE-D-20-12693

Emergency department visits among patients transported by law enforcement officers

PLOS ONE

Dear Dr. Rosen,

Thank you for submitting your manuscript to PLOS ONE. After careful consideration, we feel that it has merit but does not fully meet PLOS ONE’s publication criteria as it currently stands. Therefore, we invite you to submit a revised version of the manuscript that addresses the points raised during the review process.

We look forward to receiving your revised manuscript.

Kind regards,

Leo Beletsky

Academic Editor

PLOS ONE

Additional Editor Comments:

This is an innovative and timely analysis, but it warrants substantial fleshing out and reframing before being considered for publication. In line with the substantive and framing insights offered by the Reviewers, the authors would be well-served in situating this manuscript within the broader discourse about the role of law enforcement in responding to mental health and other health crises.

Reviewers' comments:

Reviewer's Responses to Questions

**Comments to the Author**

1. Is the manuscript technically sound, and do the data support the conclusions?

Reviewer #1: Yes

Reviewer #2: Yes

2. Has the statistical analysis been performed appropriately and rigorously? 

Reviewer #1: Yes

Reviewer #2: Yes

3. Have the authors made all data underlying the findings in their manuscript fully available?

Reviewer #1: No

Reviewer #2: Yes

4. Is the manuscript presented in an intelligible fashion and written in standard English?

Reviewer #1: Yes

Reviewer #2: Yes

5. Review Comments to the Author

Reviewer #1: This is an interesting analysis that sets out to describe frequency and patterns of law enforcement transport to emergency departments in North Carolina. This is a very worthwhile analysis to undertake, not only for its own relevance to contemporary issues of public health and public safety but for the research this study indicates clearly needs to be undertaken in order to optimize health care delivery and the treatment of individuals with severe mental health concerns and/or who are subject to involuntary commitment in the state.

My biggest complaint about this submission is that it doesn't quite provide enough information. I am of the strong opinion that more data than the authors provide - and which the authors certainly possess - should be included. I have elaborated these data, which should be included in the manuscript, immediately below:

1) The authors note in the footnote below figure 1 that IVC calls and mental health calls are not exclusive. First, this observation absolutely must be made in the main body of the text, not only in the footnote of a figure. Second, the authors should indicate the degree to which these two types of cases do and do not overlap. I understand why they drew up the figure the way they did. This makes sense and I have no problem with it. But in the body of the text, the reader should be informed of the # and % of MH calls that were not IVC calls, the # and % of IVC calls that were not MH calls, and the number of calls that were BOTH. Understanding the extent and shape of this overlap is essential for interpreting the data presented in this manuscript.

2) Nowhere in the estimations of the prevalence of law enforcement transport do the authors indicate what percentage of all emergency personnel transports are conducted by law enforcement. Ostensibly, the denominator necessary for making this comparison (total number of EMS transports, most likely) are available in the NC DETECT dataset. It is hard to know how to interpret the prevalence rates of LE transport in urban and rural areas without also having an understanding of how often LE are transporting when EMS might have otherwise been called upon to make the transport themselves, had they been available. These numbers should be calculated and presented in the text.

3) The authors present the reader with the overall gender and age breakdown of individuals transported by LE for any reason on the bottom of Pg 7. But I am more interested to know what the demographic characteristics are of individuals transported for specific concerns. I believe it is of particular interest to public health and law enforcement researchers to know (a) the demographics of patients transported for IVC, (b) whether those demographics are significantly different between rural and urban areas, and (c) whether those demographics differ significantly from the general population. This same analysis is also worth doing for MH calls, overdose, and traumatic injury.

4) Is no data on race or ethnicity available? If so, it must be included in the calculations requested above. If not, please state so clearly and offer a brief explanation of why (i.e. "it's not record in DETECT" or "it's not standardized and the data is muck" or whatever).

With these additions, this manuscript will make a much more robust contribution to literature.

Minor comments about the draft are below:

1) line 45: For non US readers, please briefly define what categories of professional title are included in the category "law enforcement officer." I assume it is sworn police officers, sworn sheriff's deputies, state patrol, hwy patrol...anything else?

2) lines 69-70: Please clarify when stating the aims of the study that the authors are seeking to understand prevalence etc. *in North Carolina*.

3) lines 78-79: The authors describe the time frame (2009-2016) as the most recent data available. That 2016 is the most recent year of data available makes sense. But what does recent availability have to do with data from 2009? Why doesn't the data go back further? Is it available in this database? Please explain.

4) line 81: What do the authors mean by "when appropriate"? When, pray tell, is appropriate?

5) line 83: were census estimates obtained for each year under consideration (2009-2016, inclusive)? If so, please say so. If not, please explain.

6) line 89: On what do the authors base their confidence that the transport code of "walk-in following non-ambulance law enforcement transport" is overwhelmingly inclusive of all ED arrivals recorded as LE transports? As a reader who is not personally familiar with this dataset, I need some more detail to understand why this is sufficient. Are there only a few codes that can be chosen from? Is this code strictly standardized across all NC hospitals? Was there a review of any number of selected or discarded transport records to check for validity?

7) line 96: Please unpack how NCHS categories were collapsed (which old categories into which new ones) and justify the collapsing. How does this help us better understand the data?

8) lines 101-104: Could the authors do us the kindness of actually informing us of what these 13 major and 10 substituent clinical categories are? This comment in the methods section is otherwise totally uninformative.

9) lines 115-117: What does "based on" mean? Is this an exhaustive list of all codes used to indicate IVC cases identified in the review of that subset of 1000 records? We need more detail about how this list of search terms was come to in order to have any faith in its validity.

10) line 130: What do the authors mean by "overall"? Is this meant to indicate a comparison between 2009 and 2016? This is not clear.

11) lines 141-144: This may simply be a difference of disciplinary norms. If so, the authors are free to disregard this comment. But I do not think of "statistically different" as standard language for describing these findings. I would suggest replacing this term with language like "the difference was/was not statistically significant."

12) lines 156-157: Here, again, as mentioned at the very beginning of this review, the authors should reiterate the proportion of arrivals that were iVC alone, the proportion of calls that were MH alone, and the promotion that were both.

13) lines 162-163: What were the causes of death in these IVC patients? This seems a terribly important detail to include.

14) lines 217-222: I may be off the mark here, but shouldn't an increase in the number of MH tranports to EDs be an indicator that CIT programs are effective at diverting people with severe (and persistent, perhaps) mental health issues away from the criminal justice system? Where are these individuals meant to be diverted to? Because I am struck that counties with the highest number of MH transports also appear to be the counties with the lowest levels of CIT training. This reflects really poorly on CIT does it not?

15) lines 217-222: on a separate, but no less important note, the authors are presenting new data in the discussion section. This observation, in my view, needs to be moved to the results section.

16) lines 229-230: It is not clear to me how allowing IVC patients to be transported by entities other than law enforcement responds in anyway to delays in behavioral and mental health care. Are the delays caused but he patients waiting around for a cop to pick them up? That seems unlikely.

17) lines 233-235: The authors are again presenting data for the very first time in the discussion section. This comparison needs to be made in detail in the results section.

18) lines 263-264: There seems to be a run-on sentence here or perhaps a word missing.

19) line 272: What do the authors mean by "given their nature"? The meaning is not clear. I suggest they be more explicit rather than insinuate.

20) line 302: if Ms. DiRosa was involved in manuscript preparation, she should be listed as an author.

Reviewer #2: This is an important paper on an understudied and timely topic. Some major and minor points:

Major:1) In the introduction, there is a mention of NC policy re: law enforcement transport, but the intro would benefit from more in depth information relative to how unique the NC policies on law enforcement transport are; 2) In the discussion the authors allude to the fact that officers may not have medical training that equips them to properly deal with many folks they encounter who may need medical transport. In the context of a growing call to defund police and divert resources toward social and public health services, I would ask the authors to consider going one step further and speculate whether other trained public health professionals would be better equipped to respond to these types of medical emergencies. The authors do not have to belabor the point, but it should certainly be raised that perhaps law enforcement are not the right agency to respond.

Minor: 1) male and female seem switched in table 1; 2) there is no fifth limitation (but there is a sixth).

6. PLOS authors have the option to publish the peer review history of their article (what does this mean?). If published, this will include your full peer review and any attached files.

Reviewer #1: No

Reviewer #2: No

---

## [Author Response · Author response to Decision Letter 0]

26 Oct 2020

PONE-D-20-12693

Emergency department visits among patients transported by law enforcement officers

PLOS ONE

We greatly appreciate the academic editor and reviewers’ efforts in reviewing our manuscript during this challenging time, and we’re pleased that the analysis was found to be “innovative and timely.” The editor and reviewers have provided many useful suggestions to improve the manuscript. Below we provide their comments followed by our responses marked by bullet points. 

Academic editor: 

This is an innovative and timely analysis, but it warrants substantial fleshing out and reframing before being considered for publication. In line with the substantive and framing insights offered by the Reviewers, the authors would be well-served in situating this manuscript within the broader discourse about the role of law enforcement in responding to mental health and other health crises.

• We now provide greater context for the role of law enforcement in addressing mental health issues. However, the majority of transports are not related to mental health and many transports may have been conducted by law enforcement as a matter of convenience rather than a response to a crisis. We think that it is important to retain this wholistic view of transports while emphasizing the impact of those related to mental health. 

Reviewer #1: This is an interesting analysis that sets out to describe frequency and patterns of law enforcement transport to emergency departments in North Carolina. This is a very worthwhile analysis to undertake, not only for its own relevance to contemporary issues of public health and public safety but for the research this study indicates clearly needs to be undertaken in order to optimize health care delivery and the treatment of individuals with severe mental health concerns and/or who are subject to involuntary commitment in the state.

• We appreciate the reviewer’s particular interest in mental health and involuntary commitment. We also think that these issues are of great importance and have devoted more of the paper to addressing these issues. At the same time, we believe that given the lack of data regarding law enforcement’s role in medical transports, it is important that we capture the full breadth of transports that are occurring. 

My biggest complaint about this submission is that it doesn't quite provide enough information. I am of the strong opinion that more data than the authors provide - and which the authors certainly possess - should be included. I have elaborated these data, which should be included in the manuscript, immediately below:

1) The authors note in the footnote below figure 1 that IVC calls and mental health calls are not exclusive. First, this observation absolutely must be made in the main body of the text, not only in the footnote of a figure. 

• We now address this point in the methods sections (3rd to last paragraph of the section). 

Second, the authors should indicate the degree to which these two types of cases do and do not overlap. I understand why they drew up the figure the way they did. This makes sense and I have no problem with it. But in the body of the text, the reader should be informed of the # and % of MH calls that were not IVC calls, the # and % of IVC calls that were not MH calls, and the number of calls that were BOTH. Understanding the extent and shape of this overlap is essential for interpreting the data presented in this manuscript.

• We now present the extent that MH and IVCs overlap: “Across the study period, IVC accounted for 22.6% (30,800/136,240) of all law enforcement ED transports and 37% (21,633/58,694) of all law enforcement ED transports for which the primary diagnosis was a mental health condition; conversely, 70% (21,633/30,800) of all IVC transports had a mental health condition listed as the primary diagnosis.”

2) Nowhere in the estimations of the prevalence of law enforcement transport do the authors indicate what percentage of all emergency personnel transports are conducted by law enforcement. Ostensibly, the denominator necessary for making this comparison (total number of EMS transports, most likely) are available in the NC DETECT dataset. It is hard to know how to interpret the prevalence rates of LE transport in urban and rural areas without also having an understanding of how often LE are transporting when EMS might have otherwise been called upon to make the transport themselves, had they been available. These numbers should be calculated and presented in the text.

• We have modified our text to clarify that we were not in possession of the entire NC DETECT database but rather only acquired the subset of records for which the transport mode was coded as “Walk-in following nonambulance, law enforcement transport.” As a result, we are unable to estimate ED transport rates by ambulance below the level of the state. Nevertheless, to place our findings in context, we now cite a 2009 report stating that there were over 4 million ED visits—including more than 550,00 by ambulance. 

• We agree with the reviewer’s point that additional information regarding EMS transports could provide further context. In particular, it would be useful to know EMS capacity within a county and more granularly, instances when law enforcement transports occurred when EMS units were otherwise occupied/unavailable. However, after speaking with state EMS leaders and county EMS experts, it is clear that valid data are not available. Nevertheless, in the future, we plan to seek new methods (e.g. surveys of county EMS agencies) to provide greater context to these law enforcement transports. 

3) The authors present the reader with the overall gender and age breakdown of individuals transported by LE for any reason on the bottom of Pg 7. But I am more interested to know what the demographic characteristics are of individuals transported for specific concerns. I believe it is of particular interest to public health and law enforcement researchers to know (a) the demographics of patients transported for IVC, (b) whether those demographics are significantly different between rural and urban areas, and (c) whether those demographics differ significantly from the general population. This same analysis is also worth doing for MH calls, overdose, and traumatic injury. 

• We present the age and gender distribution for the general population, all persons transported by law enforcement, and those transported for mental health, IVCs, overdose and wounds. We stratified by urbanicity and conducted tests of chi-square to examine whether differences across urbanicity strata achieve statistical significance. We also provide age and gender distributions for the general population, stratified by urbanicity. We did not test for differences with the general population because 1) the estimates are clearly different and 2) given the large size of the general population, even minute differences between the general population and the transported population would result in statistically significant results. We note that while gender and age often differed modestly by urbanicity within each condition, they were relatively similar across conditions with the exception of Wounds in which the vast majority occurred among men. 

4) Is no data on race or ethnicity available? If so, it must be included in the calculations requested above. If not, please state so clearly and offer a brief explanation of why (i.e. "it's not record in DETECT" or "it's not standardized and the data is muck" or whatever).

• That’s correct, regrettably those variables were not available. After describing the variables available in the dataset, we now state that variables for race/ethnicity were not collected by NC DETECT.

With these additions, this manuscript will make a much more robust contribution to literature.

Minor comments about the draft are below:

1) line 45: For non US readers, please briefly define what categories of professional title are included in the category "law enforcement officer." I assume it is sworn police officers, sworn sheriff's deputies, state patrol, hwy patrol...anything else?

• Unfortunately, specific titles were not included in the nationally representative survey of US residents from which these estimates are based. The survey asks participants about their interaction with ‘police,’ but provides no definition of what job positions this term encompasses. (https://www.bjs.gov/content/pub/pdf/cpp15.pdf). 

2) lines 69-70: Please clarify when stating the aims of the study that the authors are seeking to understand prevalence etc. *in North Carolina*.

• Thank you, we now specify “North Carolina.” 

3) lines 78-79: The authors describe the time frame (2009-2016) as the most recent data available. That 2016 is the most recent year of data available makes sense. But what does recent availability have to do with data from 2009? Why doesn't the data go back further? Is it available in this database? Please explain.

• Great point. We have removed that line and now state that “at the time of our data request, 2016 was the most recent year with available data.” We could have possibly retrieved earlier data when making our data request, but we felt that eight years was an adequate time span to examine trends and we were cognizant that NC DETECT personnel preferred that we take a more parsimonious approach in that data we obtained. 

4) line 81: What do the authors mean by "when appropriate"? When, pray tell, is appropriate?

• We are now more explicit in our description and for interested readers, we provide a additional reference describing the process. “When hospitals use different coding systems for key variables, standardization of those variables is conducted using Data Elements for Emergency Department Systems. Additional coding details are described elsewhere (15).” 

5) line 83: were census estimates obtained for each year under consideration (2009-2016, inclusive)? If so, please say so. If not, please explain.

• That is correct. The sentence now reads, “We also obtained US census estimates for NC counties for years 20009-2016, inclusive [16].”

6) line 89: On what do the authors base their confidence that the transport code of "walk-in following non-ambulance law enforcement transport" is overwhelmingly inclusive of all ED arrivals recorded as LE transports? As a reader who is not personally familiar with this dataset, I need some more detail to understand why this is sufficient. Are there only a few codes that can be chosen from? Is this code strictly standardized across all NC hospitals? Was there a review of any number of selected or discarded transport records to check for validity?

• We carefully note in the limitations section that information on transport code could be miscoded and in fact data related to transportation mode is commonly missing (~25%). We are unaware of efforts to validate this particular variable. Nevertheless, the reviewer is correct that there are relatively few possible transport categories, and we chose the code based on the guidance from NC DETECT leadership. 

7) line 96: Please unpack how NCHS categories were collapsed (which old categories into which new ones) and justify the collapsing. How does this help us better understand the data?

• As we now discuss, collapsed categories represented health conditions that appeared infrequently in our data and were of modest substantive interest. 

8) lines 101-104: Could the authors do us the kindness of actually informing us of what these 13 major and 10 substituent clinical categories are? This comment in the methods section is otherwise totally uninformative.

• We apologize that this was not clearer—we now explain that the categories are those presented in table 2; given the large number of categories (i.e. 23), we thought it best not to list each category in the text. 

9) lines 115-117: What does "based on" mean? Is this an exhaustive list of all codes used to indicate IVC cases identified in the review of that subset of 1000 records? We need more detail about how this list of search terms was come to in order to have any faith in its validity.

• We provide additional detail in the text. To be clear, we did not examine a subset of 1000 records, but rather examined the 1000 most common chief complaints. As we now describe, these 1000 most common chief complaints account for the chief complaints that are listed in about 70% of all observations (~96,000 records). Addressing the reviewer’s question about whether our list was exhaustive, we state that “we identified all text variations indicative of IVCs” among the 1000 most common chief complaints. As described in our original text, after creating our query, we then reviewed the chief complaints identified by our query and verified that that they indeed reflected IVCs. As we noted in the limitations section, it is possible that we missed instances when IVC was not clearly documented in the chief complaint text, in which case our estimates may be conservative. However, scanning over the tens of thousands of remaining chief complaints (among the 30% of remaining records), we were not able to identify any instances in which our query failed to identify an IVC. 

10) line 130: What do the authors mean by "overall"? Is this meant to indicate a comparison between 2009 and 2016? This is not clear.

• That interpretation is correct. We now state: “The annual number of visits increased from 2009 (n=13,412) to 2012 (n=17,938), declined in 2013 (n=16,148), and increased through 2016 (n=21,949) for an overall increase from 2009 to 2016 of 61%.

11) lines 141-144: This may simply be a difference of disciplinary norms. If so, the authors are free to disregard this comment. But I do not think of "statistically different" as standard language for describing these findings. I would suggest replacing this term with language like "the difference was/was not statistically significant."

• We have adopted the suggested language: “The differences in the distribution of rates between Rural counties and both Small to Medium counties (p < 0.05) and Large Metro counties (p < 0.05), were statistically significant, but the difference in the distributions of rates for Small to Medium counties compared to those of Large Metro counties were not statistically significant (p = 0.26).

12) lines 156-157: Here, again, as mentioned at the very beginning of this review, the authors should reiterate the proportion of arrivals that were IVC alone, the proportion of calls that were MH alone, and the promotion that were both.

13) lines 162-163: What were the causes of death in these IVC patients? This seems a terribly important detail to include.

• We do not have cause of death, but now provide the primary diagnosis for the top 90% (the bottom 10% are not reported b/c we are prohibited by our DUA from reporting estimates for which the n <5). 

14) lines 217-222: I may be off the mark here, but shouldn't an increase in the number of MH tranports to EDs be an indicator that CIT programs are effective at diverting people with severe (and persistent, perhaps) mental health issues away from the criminal justice system? 

• With the current data, it is not possible to conclude that an increase in MH transports is an indicator that CIT programs are effective; more information would be needed about 1) population at risk for justice involvement vs. diversion and 2) concomitant reductions in justice system involvement (i.e. arrests, conviction on criminal charges). 

Where are these individuals meant to be diverted to? 

• As stated, to area local mental health facilities. Depending on the circumstances, this could be outpatient or inpatient services. 

Because I am struck that counties with the highest number of MH transports also appear to be the counties with the lowest levels of CIT training. This reflects really poorly on CIT does it not?

• We originally abstracted the county-level % of CIT trained officers from a map of the state which was published online. Since our submission, we have received the original data/data collection forms and found several possible instances where the data may conflict with info presented on the map. Resolving these ambiguities was beyond the scope of this project, so we have dropped the comparison in the discussion. Regardless of county level CIT data, we now note that we do not have information on the extent to which officers who transported behavioral health patients had been CIT trained. Pending future funding, we plan to pursue more focused analyses examining the impact of CIT at the county level and individual level. 

15) lines 217-222: on a separate, but no less important note, the authors are presenting new data in the discussion section. This observation, in my view, needs to be moved to the results section.

• Per above, we have removed the comparison. 

16) lines 229-230: It is not clear to me how allowing IVC patients to be transported by entities other than law enforcement responds in anyway to delays in behavioral and mental health care. Are the delays caused but he patients waiting around for a cop to pick them up? That seems unlikely.

• We fully agree. We now provide more information about how the legislation, which seeks to diminish wait times and may also reduce the rates of law enforcement IVC transports. 

17) lines 233-235: The authors are again presenting data for the very first time in the discussion section. This comparison needs to be made in detail in the results section.

• With respect, we disagree. The comparison is putting our results into context with estimates for the general population based on the existing literature. This does not represent new analysis. We think that most appropriate section to provide this context is the discussion. 

18) lines 263-264: There seems to be a run-on sentence here or perhaps a word missing.

• Thank you. We have modified the passage: “Law enforcement and ED staff often work collaboratively to secure patient and staff safety. However, conflicting priorities can emerge, particularly around the use of restraints, patient confidentiality, and evidence collection via clinical laboratories or “body cams” [28].”

19) line 272: What do the authors mean by "given their nature"? The meaning is not clear. I suggest they be more explicit rather than insinuate.

• Thank you. We have modified the passage to be more explicit: “Nevertheless, we speculate that the presence of law enforcement officers with a patient and officers’ communication of patients’ security needs to hospital staff likely result in law enforcement transports being more comprehensively and accurately recorded than other transport modes (e.g. walk-ins).

20) line 302: if Ms. DiRosa was involved in manuscript preparation, she should be listed as an author.

• We appreciate the advocacy on Ms. DiRosa’s behalf. She was incredibly helpful in formatting an early version of our manuscript (e.g. converting xlsx tables to ms word tables), but she did not contribute to the intellectual content of the article. In accordance with authorship guidelines, we believe that it is most appropriate for her to be listed in the acknowledgements section but not as an author. 

Reviewer #2: This is an important paper on an understudied and timely topic. Some major and minor points:

Major:1) In the introduction, there is a mention of NC policy re: law enforcement transport, but the intro would benefit from more in depth information relative to how unique the NC policies on law enforcement transport are; 

• After consulting with several experts, we were unable to identify documentation describing differences in law enforcement transports in NC as compared to other states. However, to the reviewer’s point, in the background, we address the fact that in NC and in other states, decisions around transport are often made locally; we cite the experts that we spoke with supporting this point. 

2) In the discussion the authors allude to the fact that officers may not have medical training that equips them to properly deal with many folks they encounter who may need medical transport. In the context of a growing call to defund police and divert resources toward social and public health services, I would ask the authors to consider going one step further and speculate whether other trained public health professionals would be better equipped to respond to these types of medical emergencies. The authors do not have to belabor the point, but it should certainly be raised that perhaps law enforcement are not the right agency to respond.

• We now address this point. 

Minor: 1) male and female seem switched in table 1; 2) there is no fifth limitation (but there is a sixth).

• Thank you for pointing out that the N’s for male and female were switched. We have corrected this error.

• Thank you, we have corrected the numbering error in the limitations section.

---

## [Decision Letter · Decision Letter 1]

15 Dec 2020

Emergency department visits among patients transported by law enforcement officers

PONE-D-20-12693R1

Dear Dr. Rosen,

We’re pleased to inform you that your manuscript has been judged scientifically suitable for publication and will be formally accepted for publication once it meets all outstanding technical requirements.

Kind regards,

Jim P Stimpson, PhD

Academic Editor

PLOS ONE

Additional Editor Comments (optional):

Reviewers' comments:

Reviewer's Responses to Questions

**Comments to the Author**

1. If the authors have adequately addressed your comments raised in a previous round of review and you feel that this manuscript is now acceptable for publication, you may indicate that here to bypass the “Comments to the Author” section, enter your conflict of interest statement in the “Confidential to Editor” section, and submit your "Accept" recommendation.

Reviewer #1: All comments have been addressed

2. Is the manuscript technically sound, and do the data support the conclusions?

Reviewer #1: Yes

3. Has the statistical analysis been performed appropriately and rigorously? 

Reviewer #1: Yes

4. Have the authors made all data underlying the findings in their manuscript fully available?

Reviewer #1: Yes

5. Is the manuscript presented in an intelligible fashion and written in standard English?

Reviewer #1: Yes

6. Review Comments to the Author

Reviewer #1: I commend the authors on their hard work with this manuscript. This version is substantially improved and, after reading it through a few times, I only have 2 extremely minor comments, both of which can be handled in proofing. First, I suggest adding a citation for the reference to Daniel Prude's death from police violence, just so readers have a source to go to for more info or, alternatively, a date and location for his death. Second, The first instance of "North Carolina" at the end of the introduction should be followed with the acronym "(NC)" that is used throughout the rest of the manuscript.

This is a really lovely piece and, with the additional detail the authors provided in the revision, is going to make an extremely valuable contribution to the literature. Kudos to all who worked on this.

7. PLOS authors have the option to publish the peer review history of their article (what does this mean?). If published, this will include your full peer review and any attached files.

Reviewer #1: No

---

## [Editor Report · Acceptance letter]

21 Dec 2020

PONE-D-20-12693R1 

Emergency department visits among patients transported by law enforcement officers 

Dear Dr. Rosen:

I'm pleased to inform you that your manuscript has been deemed suitable for publication in PLOS ONE. Congratulations! Your manuscript is now with our production department. 

Kind regards, 

on behalf of

Dr. Jim P Stimpson 

Academic Editor

PLOS ONE